# Assessment of Mini-Tablets Coating Uniformity as a Function of Fluid Bed Coater Inlet Conditions

**DOI:** 10.3390/pharmaceutics13050746

**Published:** 2021-05-18

**Authors:** Magdalena Turk, Rok Šibanc, Rok Dreu, Maja Frankiewicz, Małgorzata Sznitowska

**Affiliations:** 1Department of Pharmaceutical Technology, Medical University of Gdansk, Hallera 107, 80-416 Gdansk, Poland; m.czajkowska@gumed.edu.pl (M.T.); maja.frankiewicz@gumed.edu.pl (M.F.); 2Department of Pharmaceutical Technology, University of Ljubljana, Aškerčeva cesta 7, 1000 Ljubljana, Slovenia; sibanc@hhu.de (R.Š.); rok.dreu@ffa.uni-lj.si (R.D.); 3Institute of Pharmaceutics and Biopharmaceutics, Heinrich-Heine-University, Universitätsstr. 1, 40225 Düsseldorf, Germany

**Keywords:** mini-tablets, coating, thickness, color analysis, fluid bed system, film uniformity

## Abstract

This study concerned the quality of mini-tablets’ coating uniformity obtained by either the bottom spray chamber with a classical Wurster distributor (CW) or a swirl distributor (SW). Mini-tablets with a diameter of 2.0, 2.5, and 3.0 mm were coated with hypromellose using two different inlet air distributors as well as inlet airflow rates (130 and 156 m^3^/h). Tartrazine was used as a colorant in the coating layer and the coating uniformity was assessed by spectrophotometric analysis of solutions obtained after disintegration of the mini-tablets (n = 100). Higher uniformity of coating material distribution among the mini-tablets was observed in the case of SW distributor, even for the biggest mini-tablets (d = 3.0 mm), with an RSD no larger than 5.0%. Additionally, coating thickness was evaluated by colorimetric analysis (n = 1000), using a scanner method, and expressed as a hue value. A high correlation (R = 0.993) between inter-tablet variability of hue and UV-Vis results was obtained. Mini-tablets were successfully coated in a fluid bed system using both a classical Wurster distributor as well as a swirl generator. However, regardless of the mini-tablets’ diameter, better film uniformity was achieved in the case of a distributor with a swirl generator.

## 1. Introduction

Fluid bed film coating is a commonly used method for small particle coating (pellets, granules, crystals). The choice of different types of polymers enables not only active substance protection against the environment or taste masking [1], but also modification of the release rate, such as in enteric [2] or sustained release [3] products. Fluid bed coating can be classified as a single-pot process, where the cores are coated and dried in the same unit, eliminating the product transfer and minimizing the risk of cross-contamination.

During the fluid bed coating process, the air/gas enters through the distributor into the coating chamber and creates a particle suspension with distinctive characteristics, which is wetted by a coating mixture and simultaneously dried in the air/gas stream. Such a system maximizes the exposure of the core’s surface to the coating mixture and allows the coated product to dry quickly. The main challenge related to using this technology is the selection of the proper process conditions that lead to a robust product. To facilitate the selection of proper process conditions, the statistical tools known as experimental design may be used. These highly organized experimental plans allow for identifying the most significant process parameters that have an impact on the quality of the product and allow the optimization of these parameters [4]. Although familiar processes may be better understood and optimized by statistical tools, a very important process for improving fluid bed coating is developing new technological solutions based on construction modification, which could eliminate observed design defects.

As a result of improving the construction of fluid bed systems, several coating methods were developed. These methods can be classified by the direction of coating dispersion spraying (top, bottom, tangential) or the design of the air distributor (screen, Wurster, rotor) [5]. The air distributor located at the bottom part of the chamber is intended to support stable fluidization without creating “dead zones”, where permanent or transient non-fluidized regions may occur. The well-designed distributor should also function without the occurrence of gas-particle flow plugging or failures, with minimum core attrition, and avoiding losses in the mass of the bed during the process initiation and termination [6].

The classical bottom spray Wurster chamber has been well-known since the 1960s [7], and so a thorough investigation of its impact on particle coating quality and uniformity was performed. Uniform particle film formation in the Wurster chamber allows for the widest application range, including not only a taste-masking coating but also modified release, which is not possible in the case of top spray [8]. However, the conventional Wurster chamber is not perfect in terms of particle size and preferential coating, and some modifications can be applied. A new, patented swirl distributor (SW, swirl Wurster) can be combined with the design of a classical Wurster chamber [9]. It was demonstrated that a modified chamber with a SW may result in improved per-particle coating uniformity and a higher coating yield compared with a classical Wurster (CW). The main difference between the two systems is the air distributor, which in CW is perforated and flat, while in SW it is curved with a thicker circular center segment in the form of inclined gaps (Figure 1). As a result, the SW distributor reduces dead zones performed often near chamber walls and creates a distinctive particle flow pattern within a draft tube [10,11]. To compare the effect of air distributors, different size fractions of pellets in the range from 300 to 1000 µm were coated in SW and CW. The results indicate that using SW, it is possible to improve the coating process yield with a more uniform coating layer, regardless of the pellets’ size fraction [12].

Mini-tablets, thanks to their small size (up to 3–4 mm), offer a wide range of applications, especially in pediatric and geriatric populations [13,14,15,16]. They may be coated either in drum coaters, as with traditional tablets [17], or in a fluid bed system [18,19].

Film thickness and morphology are key factors controlling release profiles, assuring API stability, and taste masking. By knowing the surface area of the total particles, the amount of processed coating mixture and dry film density, the film thickness can be estimated. Unfortunately, this is only a theoretical value, because the loss of coating material during the process (attrition, wall adherence, spray drying) may fundamentally influence the real film thickness. Thus, precise measurements based on the preliminary coating process are required to include coating yield in the calculation of the required amount of coating dispersion [20]. Assessment of film thickness and its uniformity can be performed by many different techniques where sample preparation is required: optical, fluorescence, and SEM microscopy [21,22]. Non-invasive methods including Raman [23,24] or NIR [25,26] spectroscopy, terahertz pulsed imaging [27,28], dynamic image analysis [29], and optical coherence tomography [30,31] have undoubted advantages.

Achieving uniform polymer deposition across the surface of spherical pellets seems to be easier than on particles with sharp edges, such as mini-tablets. However, compared with pellets, the coating of mini-tablets has more advantages than disadvantages [32]. As they are produced by compression, mini-tablets have completely uniform shapes and very narrow size distribution, a smoother surface, and high mechanical resistance, which leads to a robust coating process and may result in more uniform film thickness [33]. Consequently, mini-tablets need less coating material than pellets to achieve the same film thickness and to ensure similar dissolution profiles [32]. The reduction in the amount coating materials used during the process results in a shorter process time and hence generates savings. Although several publications have already reported on this process [34,35,36], there is still little knowledge about fluid bed coating of mini-tablets. Mini-tablets are a novel and promising formulation, enabling the introduction of modified release medicinal products into pediatric therapy [37]. Moreover, because of their small size and easier production [38], they may represent a relevant replacement for pellets commonly used as a multiparticulate form of the drug. Therefore, research aiming to increase knowledge about the coating process of mini-tablets is an important topic in pharmaceutical technology.

In the industry, the coating quality is frequently only assessed visually, which includes observation of the color uniformity. However, color assessment should not rely on visual evaluation, because the color is a more complex structure that may be described by three main attributes: hue, value, and chroma. Hue is expressed as a distinctive quality of color, such as red, yellow, etc. Value is described as the relation of one object to another by using a reference to light and shade. Chroma is the degree of color intensity [39]. The color quality of coated pellets was tested using a computer scanner where the main interest during measurement was focused on hue. Additionally, based on scanner measurement, it was possible to assess film thickness on pellets by estimation of the particle size distribution shift [40].

The aim of the study was to perform a comparison of mini-tablets’ coating attributes achieved after coating of the cores in two fluid bed process chambers equipped with different inlet air distributors: CW and SW. Three different sizes of mini-tablets were investigated. Coating quality was evaluated mainly based on hue to determine the usefulness of the scanner method in the analysis of mini-tablets.

## 2. Materials and Methods

### 2.1. Materials

Cores: microcrystalline cellulose (Vivapur 102, JRS Pharma, Rosenberg, Germany), sodium stearyl fumarate (PRUV, JRS Pharma, Rosenberg, Germany), spray-dried lactose (Flowlac 100, Meggle Pharmaceutical, Wasserburg, Germany), colloidal silicon dioxide (Aerosil 200, Evonik Industries AG, Darmstadt, Germany).

Coating: hypromellose (HPMC, Pharmacoat 606, Shin-Etsu Chemical, Tokyo, Japan), polyethylene glycol (PEG 6000, Sigma-Aldrich, Steinheim, Germany), tartrazine (Sigma-Aldrich, Steinheim, Germany).

### 2.2. Methods

#### 2.2.1. Placebo Cores Preparation

Biconvex mini-tablets (MT) with a diameter of 2.0 (MT2.0), 2.5 (MT2.5) and 3.0 mm (MT3.0) were prepared from a tablet mass consisting of Flowlac 100 (74.07%), Vivapur 102 (24.68%), Aerosil 200 (0.25%) and PRUV (1%).

Placebo MT were compressed using a rotary tablet press (Erweka RTP-D8, Heusenstamm, Germany) equipped with single or multiple (13) punches. The force applied for multiple punches to produce MT2.0 was 12 kN. For tableting with single punches, tooling compression forces of 2 (MT2.5) and 2.5 kN (MT3.0) were needed. Produced MT were tested according to the pharmacopeia procedures (European Pharmacopoeia, Ph.Eur. 10), including mass uniformity, thickness and friability. Disintegration time (n = 6) was tested with a standard apparatus (Pharmatest, Hainburg, Germany) equipped with a 1.0 mm sieve size.

The crushing force of uncoated and coated MT (n = 100) was measured by means of a non-pharmacopeial method, using a Texture Analyzer TA.XT.plus (Stable Micro Systems, Surrey, UK) with a probe (ø 6 mm) being constantly lowered (0.5 mm/s) until 30% of a MT’s diameter level was achieved.

#### 2.2.2. Coating Procedure

A 10% (*w*/*w*) aqueous coating dispersion consisting of HPMC (8%), PEG 6000 (1%) and tartrazine (1%) as a colorant was prepared. The required amount of water was heated up to 60 °C, and HPMC and PEG were partially added to the agitated water (300 rpm). Colorant was added to the solution of the polymer. The coating mixture was cooled down to room temperature with constant stirring for at least 30 min.

The coatings were performed in a bottom spray fluid bed system GPCG 1 (Glatt, Dresden, Germany) with the process chamber equipped with one of two distinct air distributors: a classical air distributor—CW (plate C, Glatt, Dresden, Germany)—or a swirl generator with a curved plate—SW (Brinox process systems, Sora, Slovenia). In each set, the Wurster partition was placed at the height of 25 mm above the distributor plate. The coating mixture was sprayed by a binary nozzle with an opening diameter of 1.0 mm.

Then, 1000 g of MT were preheated (10 min) at an inlet air temperature of 55 °C using an inlet airflow rate of 80 m^3^/h. After preheating, the cores were coated using the amount of the coating mixture calculated as sufficient to obtain a theoretical 20 µm film thickness (for calculation, the total surface area of mini-tablets and density of the coating film were included). Constant process parameters including inlet air temperature (55 °C), spraying rate (5 g/min) and pressure (2 bar) were maintained at the two different inlet airflow rates (130 or 156 m^3^/h). Six coatings were performed using CW. Additionally, three coatings with SW using an inlet airflow rate of 156 m^3^/h were tested for each MT size. After the spraying stage, MT were dried for 10 min in GPCG 1 system without changing airflow rate or temperature.

#### 2.2.3. Analysis of Coating Uniformity

Analysis of the uniformity of the amount of coating material on MT was performed based on UV-Vis measurements of the tartrazine assay in individual units. One hundred MT from each batch were analyzed using the UV-Vis method. Each MT was placed in a single vial and 4.5 mL of dihydrogen phosphate buffer (6.5 pH) was added. After disintegration, the vials were centrifuged (15 min, 3500 rpm) to separate undissolved particles. Samples were analyzed (UV spectrophotometer HP8453, Hewlett-Packard, Santa Clara, CA, USA) at a wavelength specific for tartrazine, 425 nm [41]. Based on tartrazine concentration, coating mixture composition, dry coating density (1.27 g/cm^3^) and the surface of a single MT, the average film thickness and relative standard deviation (RSD) for the tested units were calculated [42].

#### 2.2.4. Color Analysis of Mini-Tablets

At least 1000 MT were imaged using an Epson Perfection V700Photo scanner (Epson, Nagano, Japan). Scanning was performed with 48-bit color depth and a resolution of 600 dpi. All color and image corrections were disabled in the scanner software. MT were oriented horizontally on the central part of the scanner surface (72 × 296.8 mm^2^) with the scanner lid open, to obtain an image of MT with a black background (Figure 2). The color analysis was performed using an in-house OpenCV C++ program, which performed image segmentation (detection of individual MT) and color analysis. The color analysis was done in the Hue, Saturation and Value (HSV) color space. The average hue value of the central 3 × 3 pixels (127 × 127 µm^2^) of each mini-tablet was recorded and used in further analysis. The hue for red-yellow colors is defined as 60 * (G − B)/(R − B), where G is green, B is blue and R is the red color channel. The recorded hue value correlates with the mini-tablet local coating thickness [40]. For analysis of the results’ variation, the interquartile range (IQR) divided by the median was calculated, which can be expressed using percentiles: (H_75_ − H_25_)/H_50_.

## 3. Results and Discussion

### 3.1. Placebo Mini-Tablets Development

No difficulties were observed during the tableting of MT, as evidenced by high mass uniformity (RSD lower than 1.5%). Prepared MT had appropriate mechanical properties with low friability (below Ph. Eur. Limit for tablets which is 1%) and high hardness. Such mechanical quality of MT is required to prevent in-process attained surface defects, which can have a detrimental effect on applied coating quality. The characteristics of obtained MT are presented in Table 1.

### 3.2. Film Thickness Uniformity

During coating with CW using a lower inlet airflow rate (130 m^3^/h), irregularities in MT movement near the chamber walls were observed. This was particularly evident for bigger MT cores. Increasing the inlet airflow rate to 156 m^3^/h resulted in better circulation. This problem was not noticed with the SW distributor, regardless of MT diameter.

Tartrazine, a water-soluble colorant, was dissolved in the coating mixture, which ensured its homogenous distribution in the coating layer. Therefore, film thickness can be assessed based on the UV-Vis analysis of colorant, deposited on single units. For all applied coatings, regardless of the type of distributor or used inlet airflow rate, an average film thickness (20 µm) was designed based on the characteristics of cores and coating dispersion. The biggest deviations of UV-Vis-calculated average coating thicknesses (Table 2) were observed in the case of MT2.5, where the average coating thickness of batches ranged from 17.9 to 21.9 µm.

Achievement of around 20 µm film thickness (on caps, edges and around bands) in each batch was also confirmed by stereoscopic microscopy (Opta-Tech, Warsaw, Poland). An example microscopic photo of coated MT cross-section is presented in Figure 3.

However, more interesting are the RSD values of MT film thickness (Figure 4), as a good film uniformity indicator. Usage of a CW distributor with 130 m^3^/h airflow rate resulted in the highest RSD, increasing with the MT diameter (more than 19% for MT3.0). Higher film uniformity was observed for 156 m^3^/h, but still, the RSD of the largest cores exceeded 12%. The best results were obtained for SW, where RSD for MT2.0 and MT2.5 is close to 3%, while for MT3.0 it is less than 5%. Longer coating time in general resulted in a lower coating variability. The RSD of coating is inversely proportional to the square root of coating time [43]. Since all MT were coated to the same coating thickness, the coating time for smaller MT was longer, which can partially explain the better coating results for MT2.0. An extrapolation for MT2.5 and MT3.0 to the coating time of MT2.0 was performed. It was observed that the coating results still follow the same trend, which is that coating variability increases with MT size. The influence of MT size on the coating thickness RSD value also can be commented on from the perspective of the fixed gap size used in the coating experiments. The ratio between gap size and MT diameter ranged from 12.5 for MT2.0 to 8.3 for MT3.0, and this is without considering the worst-case scenario on the MT maximum Feret diameter. Gap size poses a greater obstacle to horizontal mass flow for MT3.0 than for MT2.0, which in turn reduces the total number of MT coating events per time unit (minute).

Additionally, the hardness of coated MT for samples (n = 100) from each batch was evaluated (Table 3). Only 20 µm of film thickness almost doubled MT average hardness (see also Table 1). A similar pattern as in the case of UV-Vis data for the film thickness RSD can be also deducted from the MT hardness measurements. The underlying cause for such similarity is the assumption that an individual mini-tablet with a higher coating thickness will on average exhibit a higher crushing strength. The highest hardness variability (RSD) was observed in the case of MT coated in the CW distributor at 130 m^3^/h airflow rate. Increasing the airflow rate to 156 m^3^/h resulted in more uniform hardness (RSD not more than 11%), but only a change in the distributor to SW lowered tablet hardness RSD to no more than 9%.

### 3.3. Coated Mini-Tablets Color Analysis

Color analysis was performed using a computer scanner and the hue of the individual MT was evaluated. The hue of the mini-tablet is dependent on the film coating thickness, since the applied film is evenly colored and transparent. Lower values of hue correspond to a thicker coating [40]. The median values, as well as the spread of the MT hues, can be seen in the form of a box plot (Figure 5). The interquartile ranges are in all cases larger for the CW coater when compared to the SW coater at the same process conditions. It can be also seen that MT2.5 coated with CW/130 had a large number of outliers. The larger number of outliers than in the case of smaller MT was also seen in all three coating cases with MT3.0 (regardless of distributor type). These outliers mean that there were more or stronger mini-tablet bed zones where the movement of MT was restricted. This can, for larger MT, be explained by two mechanisms. The gap was in all cases 25 mm and the largest MT had the smallest gap to size ratio, as already reported, which is important in the flow of the material to the central part of the coater, where the upward transport and coating process occurs. Secondly, the minimum fluidization velocity is also higher for larger sized MT and this is important for the mixing behavior in the annulus bed region of the coater. As the same airflow rates were used for all coating process instances, mixing was least pronounced in the case of MT3.0. The better results for the SW distributor could be attributed to the curved nature of the perforated plate in the annulus region, which promoted an even flow of MT towards the gap region.

The coating variability results obtained using UV-VIS and color analysis are summarized in Table 4. Due to a large number of outliers observed in certain analyzed cases (Figure 5), in addition to the center hue RSD, another parameter was employed to measure the coating variability. An interquartile range (IQR), normalized by the median of hue, showed a good correlation (R = 0.993) with the RSD results obtained using UV-VIS (Figure 6). However, it is important to note that the imaging color analysis has several advantages compared to the UV-VIS. The method is non-destructive, fast, and can be easily used during the analysis of a large number of samples, yielding higher statistical confidence. Based on chi-square statistics, the confidence interval is 8.780–11.617 when the standard deviation equals 10 and the number of samples is 100, and when 1000 units are measured, the confidence interval is 9.580–10.459.

Fluid bed coating of pellets using both a swirl and standard Wurster distributor were already compared by Heng et al. [44]. Pellet flows in the fluid bed coater equipped with a swirl distributor were characterized by higher Reynolds numbers (indicating higher turbulence) than standard Wurster coating. The turbulent flow of pellets in the coating chamber equipped with the swirl distributor resulted in a more uniform coating with less agglomeration and fewer defects. Thus, the benefits from using the swirl distributors are noticeable, and there have been first attempts to construct new swirl distributors using 3D printing [45].

## 4. Conclusions

Mini-tablets can be coated in a fluid bed system using both classical Wurster distributors as well as swirl generators. However, based on the results of UV-Vis analysis of the coating tracer, better film uniformity (regardless of the mini-tablets diameter) was achieved in the case of the distributor with a swirl generator. While using the same process conditions and the same process chamber, higher coating variability was always observed for bigger mini-tablets (3.0 mm diameter) than for the smaller ones (diameter of 2.5 or 2.0 mm). Increased air fluidization velocity always improved the coating uniformity of mini-tablets. It is assumed that the mixing in the annulus region, as well as the gap size, are the limiting factors that determine mini-tablet coating thickness variability. Hardness testing showed an increase in hardness for coated mini-tablets as well as good correlation with the coating thickness variability pattern. It was also demonstrated that imaging with color analysis can provide quick and reliable results concerning mini-tablet coating thickness variability. A high correlation between the inter-mini-tablet variability of hue and UV-Vis results was obtained, which indicated the usefulness of non-destructive, fast and easy color analysis, performed with a scanner, to evaluate film thickness on a large number of mini-tablets.

## Figures and Tables

**Figure 1 pharmaceutics-13-00746-f001:**
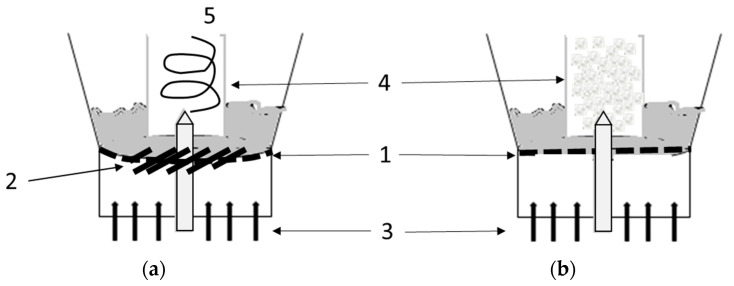
Scheme of chambers with SW (**a**) and CW (**b**) distributor: 1—curved (**a**) and flat (**b**) air distributors, 2—swirl generator, 3—air inlet, 4—draft tube, 5—particles’ movement.

**Figure 2 pharmaceutics-13-00746-f002:**
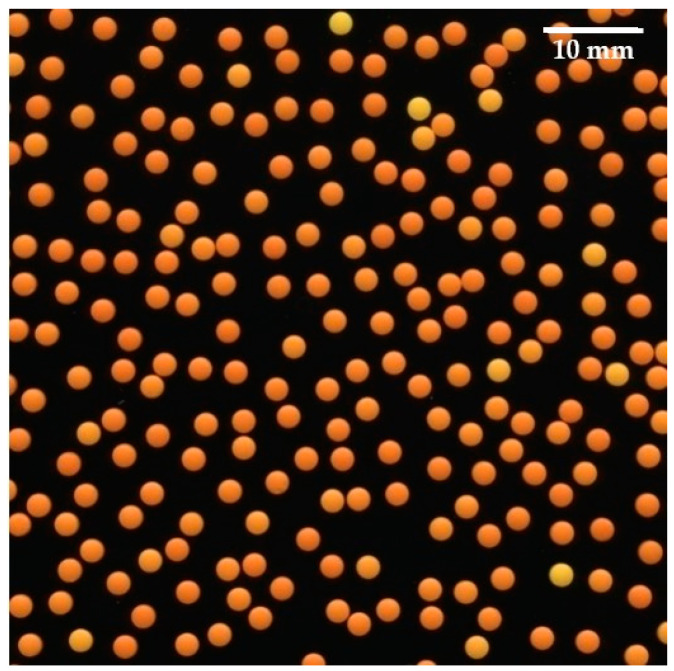
A cut-out from a scan of MT2.5 coated with CW/130.

**Figure 3 pharmaceutics-13-00746-f003:**
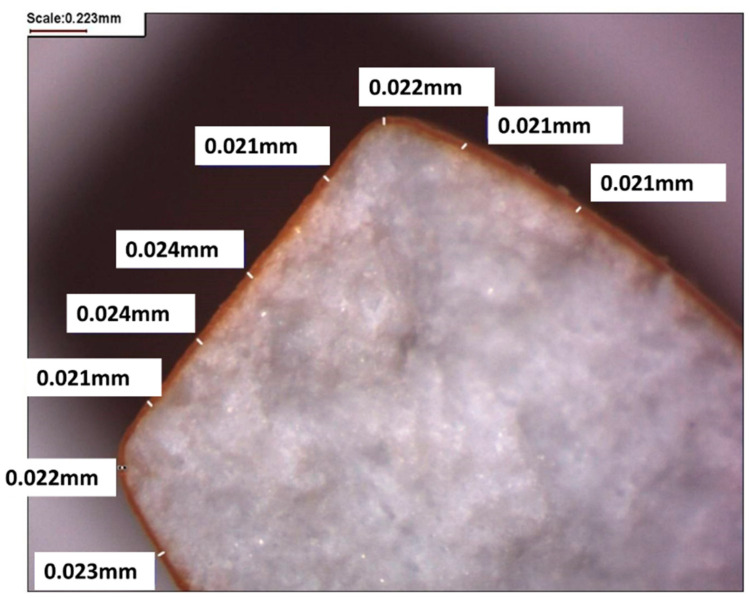
A cross-section of MT2.5 coated in the CW process chamber at 156 m^3^/h airflow rate.

**Figure 4 pharmaceutics-13-00746-f004:**
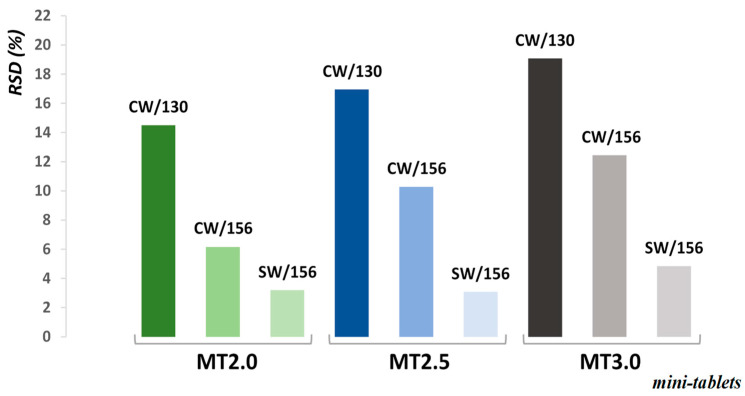
RSD (%) of film thickness for coated MT samples (distributor/airflow rate).

**Figure 5 pharmaceutics-13-00746-f005:**
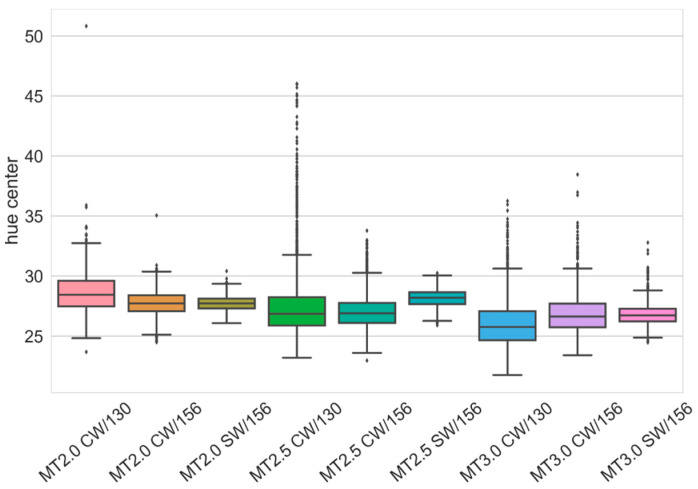
Box plot of coating hue value at the center of the mini-tablet cap face.

**Figure 6 pharmaceutics-13-00746-f006:**
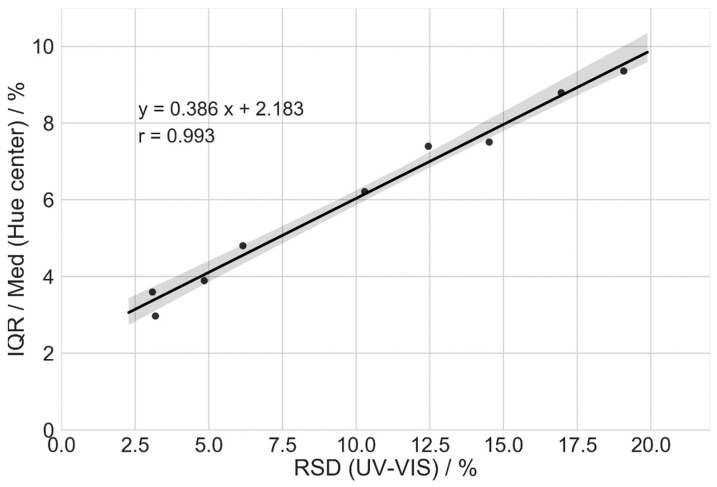
Correlation of mini-tablet center hue and coating thickness variability.

**Table 1 pharmaceutics-13-00746-t001:** Placebo mini-tablets’ characteristics.

Parameters	MT2.0	MT2.5	MT3.0
Diameter (mm)	2.0	2.5	3.0
Thickness (mm) ± RSD *	2.02 (±0.85%)	1.82 (±0.99%)	2.21 (±0.54%)
Surface of a single unit (mm^2^) ± RSD *	15.95 (±2.34%)	18.84 (±2.45%)	28.73 (±1.56%)
Mass (mg) ± RSD *	7.97 (±1.34%)	11.68 (±1.31%)	19.96 (±0.75%)
Hardness (N) ± RSD *	13.27 (±15.15%)	22.12 (±12.32%)	28.29 (±8.86%)
Friability (%)	0.3	0.21	0.15
Disintegration time (s) ± SD **	40 (±5 s)	60 (±10 s)	40 (±5 s)

* Measurement performed for n = 100; ** measurement performed for n = 6; RSD—relative standard deviation; SD—standard deviation; MT2.0, MT2.5 and MT3.0—mini-tablets with a diameter of 2.0, 2.5 and 3.0 mm, respectively.

**Table 2 pharmaceutics-13-00746-t002:** Average film thickness (µm) based on tartrazine UV-Vis analysis (n = 100) after MT coating with CW and SW distributors.

Cores	MT2.0			MT2.5			MT3.0		
Distributor	CW	CW	SW	CW	CW	SW	CW	CW	SW
Inlet airflow rate (m^3^/h)	130	156	156	130	156	156	130	156	156
Film thickness (µm)	18.7	19.9	18.4	18.9	21.9	17.9	18.8	18.2	18.3

**Table 3 pharmaceutics-13-00746-t003:** The hardness of coated mini-tablets (n = 100).

Cores	Distributor/Air Rate (m^3^/h)	Average Hardness (N)	RSD (%)
MT2.0	CW/130	33.39	14.72
	CW/156	34.36	10.96
	SW/156	33.70	9.00
MT2.5	CW/130	43.58	14.72
	CW/156	42.68	10.20
	SW/156	40.61	7.00
MT3.0	CW/130	51.04	17.67
	CW/156	53.77	9.27
	SW/156	51.57	8.10

**Table 4 pharmaceutics-13-00746-t004:** Coating variability and color variability results.

Cores	Distributor/Airflow Rate (m^3^/h)	Coating Thickness RSD UV-VIS (%)	Center Hue RSD (%)	IQR/Median Hue Center (%)
MT2.0	CW/130	14.51	6.64	7.51
	CW/156	6.15	3.68	4.81
	SW/156	3.19	2.18	2.97
MT2.5	CW/130	16.95	10.66	8.79
	CW/156	10.28	5.07	6.22
	SW/156	3.08	2.52	3.60
MT3.0	CW/130	19.08	8.02	9.36
	CW/156	12.45	6.21	7.40
	SW/156	4.84	3.21	3.89

## Data Availability

All data obtained during the study are available from the corresponding author on reasonable request.

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
