# Peer review of "Assessment of Mini-Tablets Coating Uniformity as a Function of Fluid Bed Coater Inlet Conditions"

_pharmaceutics, 2021, doi:10.3390/pharmaceutics13050746_

Round 1

Reviewer 1 Report

In the manuscript titled “Assessment of mini-tablets coating uniformity as a function of fluid bed coater inlet conditions”, Turk et al. investigated the quality of mini-tablets coating using the bottom spray chamber with classical Wurster distributor (CW) or swirl distributor (SW).

I recommended that the article can be published after addressing the comments below.

1- Line 93:  What is the relation between the tensile strength and the uniformity of the film thickness?

2- Line 161: After preheating the cores were coated until a theoretical 20 µm film thickness was obtained.

What do you mean by theoretically? Is there any formula that can be used to calculate the thickness theoretically to be 20 µm? Or this is a proposed value.

3- Line 216: the authors stated that “Tartrazine was homogeneously dispersed in the MT coating layer, therefore film -----“. What is the evidence of homogeneous dispersion?  

4- Table 4 shows that the coating thickness is less than 20 µm for most of the cases for most of the flow rate values? Could you please explain?

5- More discussion needs to be added on the effect of the distributor design on the coating mechanism and coating thickness.

6- Up to date references need to be added.  

Reviewer 2 Report

The research study by Turk et al. aimed to assess the mini-tablets coating uniformity using spectrophotometric analysis. The results showed that better film uniformity was achieved in the case of a distributor with a swirl generator compared to the Wurster distributor. In general, the study is well designed, written, and easy to follow while the topic covered is interesting to readers. I would strongly recommend accepting this manuscript for publication after a minor revision for the following comments.

Page 5, Section 3.1: Mention the limits of friability and hardness for the minitablets according to the pharmacopeia and compare the results with the limits.

Page 6, Table 1: The disintegration time results were presented with standard deviation and for other parameters, relative standard deviation has been used. I would suggest giving RSD for the disintegration time results too.

Page 6, Table 1: Add RSD values for friability results. Define MT2.0, MT2.5, MT3.0, RSD and SD at the footnotes of the table. Indicate the number of measurements in the table legend.

Page 6, Table 2: Add the standard deviation values for the film thickness and indicate the number of measurements in the table legend.

Figure 3: The numbers in the figure are difficult to read. Increase the font size of the text in the figure.

Figure 4: Correct the position of label positions.
